# Hypoxia Regulates Brown Adipocyte Differentiation and Stimulates miR-210 by HIF-1α

**DOI:** 10.3390/ijms26010117

**Published:** 2024-12-26

**Authors:** Jan Caca, Alexander Bartelt, Virginia Egea

**Affiliations:** 1Institute for Cardiovascular Prevention (IPEK), Faculty of Medicine, Ludwig-Maximilians-Universität München, 81377 Munich, Germany; jan.caca@med.uni-muenchen.de; 2German Center for Cardiovascular Research, Partner Site Munich Heart Alliance, Ludwig-Maximilians-Universität München, 80336 Munich, Germany; 3Institute for Diabetes and Cancer (IDC), Helmholtz Center Munich, German Research Center for Environmental Health, 85764 Neuherberg, Germany; 4German Center for Diabetes Research, 85764 Neuherberg, Germany; 5Chair of Translational Nutritional Medicine, Department of Molecular Life Sciences, TUM School of Life Sciences, Technical University of Munich, 85354 Freising, Germany; 6Else Kröner Fresenius Center for Nutritional Medicine, Technical University of Munich, 85354 Munich, Germany

**Keywords:** thermogenesis, brown adipocytes, hypoxia, hypoxamiRs, miRNAs, miR-210

## Abstract

MicroRNAs (miRNAs) are short sequences of single-stranded non-coding RNAs that target messenger RNAs, leading to their repression or decay. Interestingly, miRNAs play a role in the cellular response to low oxygen levels, known as hypoxia, which is associated with reactive oxygen species and oxidative stress. However, the physiological implications of hypoxia-induced miRNAs (“hypoxamiRs”) remain largely unclear. Here, we investigate the role of miR-210 in brown adipocyte differentiation and thermogenesis. We treated the cells under sympathetic stimulation with hypoxia, CoCl_2_, or IOX2. To manipulate miR-210, we performed reverse transfection with antagomiRs. Adipocyte markers expression, lipid accumulation, lipolysis, and oxygen consumption were measured. Hypoxia hindered BAT differentiation and suppressed sympathetic stimulation. Hypoxia-induced HIF-1α stabilization increased miR-210 in brown adipocytes. Interestingly, miR-210-5p enhanced differentiation under normoxic conditions but was insufficient to rescue the inhibition of brown adipocyte differentiation under hypoxic conditions. Although adrenergic stimulation activated HIF-1α signaling and upregulated miR-210 expression, inhibition of miR-210-5p did not significantly influence UCP1 expression or oxygen consumption. In summary, hypoxia and adrenergic stimulation upregulated miR-210, which impacted brown adipocyte differentiation and thermogenesis. These findings offer new insights for the physiological role of hypoxamiRs in brown adipose tissue, which could aid in understanding oxidative stress and treatment of metabolic disorders.

## 1. Introduction

Hypoxia, the biological response to inadequate oxygen supply cells and tissues, is strongly associated with various metabolic conditions such as diabetes and obesity [1]. Obesity, the pathological accumulation of white adipose tissue (WAT), significantly increases the risk of health issues such as diabetes, cardiovascular disease, and cancer [2]. In hypoxic conditions, the imbalance between oxygen supply and demand exacerbates the production of reactive oxygen species (ROS), which are key mediators of redox signaling and oxidative stress within adipose tissue [3]. In contrast, brown adipose tissue (BAT), known for its ability to generate heat and regulate energy expenditure, has been found to promote healthy adipose tissue development and improve glucose metabolism [4], thereby offering a natural protective mechanism against obesity [5]. WAT and BAT represent the two primary types of adipose tissue responsible for storing energy as triglycerides [2]. WAT releases stored triglycerides to support other tissues during energy scarcity, while BAT utilizes stored triglycerides and circulating nutrients for heat generation. The primary function of BAT as a specialized organ for non-shivering thermogenesis relies on its extensive vascularization and ample oxygen supply [5]. In BAT, heightened ROS levels during thermogenic activation may transiently support signaling pathways for energy metabolism; however, chronic hypoxia, as seen in obesity, overwhelms antioxidant defenses, disrupting redox balance and mitochondrial efficiency [3]. However, little consideration has been given to how the surge in oxygen consumption upon activation is virtually creating hypoxia. Likewise, in obesity, brown adipocytes experience increased lipid deposition, leading to the enlargement of BAT and vascular rarefaction [1,6,7]. Hypoxic conditions within adipose tissue are also associated with insulin resistance and obesity-related complications, as they promote inflammation and impair adipokine secretion. ROS-driven oxidative stress within WAT exacerbates metabolic dysfunction by promoting pro-inflammatory cytokine release and adipokine dysregulation, linking redox processes to insulin resistance and obesity complications [8]. Inhibiting hypoxia in adipocytes has been shown to improve insulin sensitivity and reduce adiposity, highlighting its potential as a therapeutic target for metabolic diseases [9,10].

Hypoxia signaling is mainly mediated by the transcription factor hypoxia-inducible factor 1-alpha (HIF-1α), responsible for the expression of most glycolytic enzymes during low oxygen levels [11]. Under normal oxygen conditions (normoxia), HIF-1α has a short lifespan and is degraded by the ubiquitin-proteasome system. The excessive ROS generated under hypoxia not only acts as a signaling molecule to stabilize hypoxia-inducible factor 1-alpha (HIF-1α) but also contributes to oxidative damage, promoting inflammation and impairing mitochondrial function [12]. However, during hypoxia, the activity of prolyl-4-hydroxylases decreases, reducing hydroxylation and degradation of HIF-1α [13,14]. HIF-1α forms a complex with the more stable HIF-1β subunit, and together they bind to specific DNA sequences called hypoxic responsive element (HRE). In recent years, next to protein-coding transcripts, hypoxia-regulated microRNAs (miRNAs), collectively referred to as hypoxamiRs, have been identified [15]. MiRNAs are single-stranded, approximately 22 nucleotide-long non-coding RNAs [16]. They play a role in post-transcriptional gene silencing by targeting messenger RNAs (mRNAs), which leads to their translational repression or decay [17]. Hypoxia-induced ROS can influence the expression of hypoxamiRs, including miR-210, by modulating HIF-1α activity, underscoring the interplay between redox dynamics and gene regulation under low oxygen conditions. Precursor miRNAs consist of two strands: miR-5p and miR-3p, which differ in sequence and therefore mRNA-transcript targets. Depending on the tissue or cell type, one miRNA strand can be selectively chosen for its function, while the other strand may undergo degradation. Alternatively, both strands can be retained and function simultaneously [18]. MiR-210 is the most extensively studied hypoxamiR [19]. In a manner similar to conventional genes, HIF-1α directly binds to an HRE located on the proximal promoter of miR-210 [20]. The conservation of this HRE site across different species, when comparing the core promoter of miR-210, underscores the importance of hypoxia and HIFs in regulating the expression of miR-210 throughout evolution [20]. However, it remains unclear whether the regulation of miR-210 during hypoxia is in the form of miR-210-5p, miR-210-3p, or both concomitantly. Given the limited focus on the occurrence of hypoxia in BAT during obesity and its effects on brown adipocyte function, this study aims to investigate the influence of hypoxia and the associated hypoxamiR, miR-210, on BAT differentiation and thermogenesis. The dual role of ROS as both signaling molecules and contributors to oxidative stress highlights the importance of tightly regulated redox processes in maintaining adipose tissue homeostasis, especially under hypoxic and obesogenic conditions. Therapeutic strategies aimed at modulating ROS levels or enhancing antioxidant defenses could help restore redox homeostasis, alleviate hypoxia-induced dysfunction in BAT, and improve metabolic health.

## 2. Results

### 2.1. Hypoxia and HIF-1α Stabilization Regulate miR-210 Expression in Brown Adipocytes

To investigate the impact of hypoxia and HIF-1α stabilization, brown adipocytes were transfected with either negative control siRNA or HIF-1α siRNA two days prior to the assay. Subsequently, the cells were exposed to normoxic (21% O_2_), and hypoxic conditions (1% O_2_) using a hypoxic incubator or in the presence of hypoxia mimetic agents as of IOX2, a selective inhibitor of the oxygen-sensitive prolyl hydroxylase domain-containing protein 2 (PHD-2), or CoCl_2_ under normoxic conditions for 24 h. These complementary approaches were chosen to comprehensively assess hypoxia-related pathways: while 1% O_2_ replicates physiological hypoxia by reducing oxygen availability, IOX2 selectively stabilizes HIF-1α without oxygen depletion, and CoCl_2_ chemically mimics hypoxia while inducing broader transcriptional responses. Immunoblot analysis revealed higher protein levels of HIF-1α under physical and chemical hypoxic induction in the cells transfected with negative control siRNA, while HIF-1α protein was absent in the HIF-1α knockdown conditions (Figure 1A). HIF-1α stabilization and subsequent nuclear translocation could also be confirmed by immunofluorescence analysis after incubation of the cells under normoxic conditions with IOX-2 or under hypoxic conditions for 12 h (Figure 1B). Consistently, mRNA levels of HIF-1α target genes such as *Vegfa*, *Serpine1*, and *Egln3* were higher under hypoxic conditions. Under the same conditions, we investigated the impact of hypoxia on the expression of miR-210 using qRT-PCR (Figure 1C). We observed that incubating brown adipocytes under hypoxic conditions resulted in a minimum 2-fold higher level of both strands of miR-210, with the effect being more pronounced for 5p (Figure 1D). In normoxia, incubating brown adipocytes with IOX-2 or CoCl_2_ also led to a more than 2-fold upregulation of both miRNA strands (Figure 1D). This up-regulation of miR-210 strands by hypoxic conditions was diminished by knocking down HIF-1α (Figure 1D). Given the pronounced induction of miR-210-5p under these conditions, further studies focused on this strand.

### 2.2. Hypoxia Impairs Brown Adipocyte Differentiation

To investigate the influence of hypoxia on brown adipogenic differentiation, we exposed immortalized WT-1 mouse brown preadipocytes to the standard protocol under both normoxic conditions (21% oxygen) and hypoxic conditions (1% oxygen) for 5 days. Microscopic analysis revealed that cells differentiated under normoxic conditions exhibited typical multilocular lipid droplets, whereas those under hypoxic conditions did not (Figure 2A). PDGFRα was also analyzed as an early brown adipocyte progenitor marker, which typically decreases during brown adipogenic differentiation [21]. Surpringly, Western blot analysis of PDGFRα demonstrated that, under hypoxic conditions, PDGFRα expression remains elevated. (Figure 2B) This effect was also observed through immunofluorescence analysis (Figure 2C). In the same fashion, the expression of the adipocyte differentiation markers *Adipoq*, *Fabp4*, and *Pparg* was remarkably downregulated under the hypoxic conditions, as shown by qRT-PCR (Figure 2D). To investigate the role of miR-210-5p in these effects, we transfected the cells with negative control miRNA (NC) or LNA miR-210-5p inhibitor (I) one day before initiating adipogenic differentiation in pre-adipocytes. MiR-210-5p inhibition negatively affects brown adipocyte differentiation and lipid storage capacity, as observed in qRT-PCR analysis (Figure 2D) and through Oil Red O staining, respectively (Figure 2E,F). Overexpression of miR-210-5p using LNA-mimics enhanced brown adipocyte differentiation under normoxic conditions and partially restored lipogenic capacity under hypoxia, as shown by Bodipy staining (Figure 2G).

### 2.3. Hypoxia Impairs the Thermogenic Capacity of Brown Adipocytes

To simulate cold-induced adrenergic receptor activation, the cells were treated with the β3-adrenergic agonist CL316243 (CL). Mouse primary brown adipocytes were transfected with either NC miRNA or LNA miR-210-5p inhibitor and treated with CL under normoxic or hypoxic conditions for 4 h. The incubation with CL resulted in higher expression levels of *uncoupling protein 1 (UCP1)* under normoxic conditions (Figure 3A). However, this effect was counteracted under hypoxic conditions. Notably, the inhibition of miR-210-5p showed no significant effect, as confirmed by qPCR analysis (Figure 3A). The counteracting effect of hypoxia on CL was only evident in the expression of the thermogenic marker *Elovl3* (Figure 3B). As *UCP1* mRNA expression does not directly correlate with its functional activity, we assessed the oxygen consumption rate (OCR) in differentiated brown adipocytes treated with the HIF-1α stabilizer IOX-2 (Figure 3C). Remarkably, IOX-2 significantly attenuated the CL-induced response, blunting β3-adrenergic receptor activation’s effects on cellular respiration, ATP-linked respiration, and maximal oxygen consumption rate (Figure 3D).

### 2.4. Adrenergic Stimuli Activate HIF-1α and Increase miR-210 Expression

To investigate the impact of adrenergic stimuli on the HIF-1α-mediated response, we induced the thermogenic program in differentiated brown adipocytes by exposing them to norepinephrine (NE) and CL for different durations (2, 4, 8, or 24 h). NE was used as a physiological agonist to activate all adrenergic receptor subtypes (β1, β2, and β3), while CL, a selective β3-adrenergic agonist, was used to specifically target β3-adrenoceptors, predominantly expressed in brown adipocytes, allowing comparison of general adrenergic and β3-specific effects. Through immunoblotting, we observed that both NE and CL increased the levels of HIF-1α protein, with NE showing a more pronounced effect (Figure 4A). Additionally, both NE and CL caused a minimum 2-fold increase in the expression of both strands of miR-210 in brown adipocytes after 24 h (Figure 4B). To investigate the potential relationship between miR-210-5p and UCP-1 expression, we conducted an experiment where brown adipocytes were transfected with either NC miRNA or an LNA inhibitor specific to miR-210-5p. Subsequently, the cells were incubated with CL (1 µM) for a duration of 6 h. Our findings from qPCR analysis (Figure 4C) indicate that the inhibition of miR-210-5p did not affect UCP1 expression. However, when different concentrations of the miR-210-5p inhibitor were used, a dose-dependent increase in UCP-1 protein levels was observed under CL stimulation, as demonstrated by immunoblotting (Figure 4D), although slight variability between the 40 nM and 80 nM treatments was noted. This minor variability in band intensity may be attributed to the semi-quantitative nature of Western blot analysis, but the overall trend of UCP-1 protein increase under adrenergic stimulation remained consistent. Besides, CL-treated samples showed increased expression of mitochondrial complexes (Figure 4D). Following these results on UCP1 protein expression, we also analyzed the oxygen consumption rate of transfected brown adipocytes. No significant differences were observed in oxygen consumption when miR-210-5p inhibition was applied (Figure 4E). Moreover, lipolysis remained unaffected by the inhibition of miR-210-5p, as shown by the quantification of glycerol concentration 6 h after incubation with either NE or CL (Figure 4F). In conclusion, adrenergic stimuli activate HIF-1α signaling and miR-210 expression.

## 3. Discussion

BAT is distinct from WAT as it primarily generates heat through thermogenesis rather than “only” storing energy. This unique function of BAT, achieved by activating UCP1 and other futile cycling mechanisms, makes it an appealing target for therapeutic interventions against metabolic disorders [22]. Here we provide evidence that hypoxic conditions impair brown preadipocyte differentiation and inhibit the thermogenic response during sympathetic stimulation. Hypoxia-induced oxidative stress, driven by an overproduction of ROS, likely exacerbates this impairment by disrupting redox signaling pathways crucial for differentiation and thermogenic activation in brown adipocytes. Excess ROS further destabilizes mitochondrial function, a cornerstone of BATs thermogenic capacity. Our findings demonstrate that miR-210 expression increases in brown adipocytes under conditions of both HIF-1α stabilization and sympathetic stimulation. While we could identify a novel involvement of miR-210-5p in the differentiation capacity of brown adipocytes under hypoxic conditions, miR-210-5p appears not to be essential to support adaptive thermogenesis in mature brown adipocytes. Although both miR-210-5p and miR-210-3p are upregulated under hypoxic conditions (Figure 1D), we specifically focused on miR-210-5p in this study due to its emerging role in regulating hypoxia-related processes. While miR-210-3p has been more extensively studied and is considered the functional strand in many cellular contexts, recent evidence suggests that miR-210-5p may also contribute to cellular responses under certain conditions [23]. The pronounced induction of miR-210-5p observed in our system led us to focus on this strand to investigate its role in brown adipocyte differentiation under hypoxic conditions.

Based on our studies, a hypoxic environment has an adverse impact on the differentiation of brown preadipocytes. In such conditions, differentiated adipocytes not only lack the typical characteristics of BAT, but they also exhibit a distinct morphology compared to undifferentiated adipocytes. This suggests that hypoxic conditions may impair adipogenic differentiation or promote alternative differentiation pathways of brown preadipocytes. The increased ROS production under hypoxic stress not only disrupts differentiation but also amplifies oxidative damage, altering cellular structures and promoting the activation of stress-responsive pathways. This could shift differentiation away from a thermogenic phenotype. This data are in consonance with previous studies showing the detrimental effects of hypoxia also on the differentiation of many other cell types, including osteocytes [24], beige adipocytes [1], cardiomyocytes [25], and myoblasts [26]. Our findings suggest that hypoxia could be critical for the maintenance of the undifferentiated precursor cell phenotypes in the stem cell niches [27,28,29]. Furthermore, alterations in the microenvironment caused by elevated insulin levels [30], oxidative stress, and reduced oxygen levels could impair the differentiation potential of pre-brown adipocytes. While transient ROS signaling can support initial cellular responses, chronic oxidative stress under hypoxic conditions may inhibit differentiation, hinder thermogenesis, and exacerbate metabolic dysfunction, ultimately affecting adipose tissue homeostasis [3].

In this study, we have also observed that inhibition of miR-210-5p reduces the ability of brown preadipocytes to differentiate, which depicts contrary effects for hypoxia and its hypoxamiR. Previous studies have already established a connection between miR-210 in white adipocyte differentiation. For example, Qin et al. conducted miRNA expression profiling during white adipocyte differentiation and identified 18 miRNAs, including miR-210, that promote adipocyte differentiation by inhibiting Wnt signaling [31]. Overexpression of miR-210 and the white adipose cell line 3T3-L1 resulted in larger cells with distinct lipid droplets, while its inhibition led to reduced adipogenesis [31]. Interestingly, miR-210 has also been shown to enhance the differentiation of mesenchymal stem cells into the osteogenic and chondrogenic lineages [32]. Moreover, miR-210 has been implicated in redox regulation by influencing mitochondrial respiration and ROS production under hypoxic conditions. Its ability to modulate oxidative stress further supports its role as a critical mediator in hypoxia-driven adipocyte differentiation [33]. In contrast to our findings, previous studies have also demonstrated that the delivery of miR-210 through exosomes hinders the process of adipose browning by affecting the FGFR-1 signaling pathway [34]. However, it is important to note that these findings were obtained in a high-altitude model. It is possible that the limitations of our experimental setup, which involved using an immortalized brown adipocyte cell line, prevented us from replicating the biological complexity and natural regulation of BAT activation or remodeling that is observed in vivo in mice or humans.

In this study, we also detected an increase in HIF-1α and miR-210 levels during sympathetic stimulation of brown adipocytes. Cold exposure is known to induce hypoxia in both brown and beige adipocytes [35,36]. This is because, in both BAT and the inguinal adipose tissue, HIF-1α stabilization results from stimulation of the activity of UCP1 in the mitochondria of the adipocytes. UCP1 uncouples oxidative phosphorylation from ATP production, resulting in increased thermogenesis but also heightened oxygen consumption. This elevated oxygen demand surpasses the local oxygen supply, creating a hypoxic microenvironment [37]. Additionally, the elevated oxygen consumption during thermogenesis drives ROS production in brown adipocytes. While ROS may initially enhance UCP1 activation, their accumulation under prolonged hypoxia disrupts mitochondrial function, contributing to brown adipose tissue dysfunction [38].

Previous research, including this, indicated that HIF-1α may negatively impact BAT respiration. Therefore, we hypothesized that inhibiting miR-210-5p could modulate the expression of uncoupling proteins. Surprisingly, the miR-210-5p inhibitor significantly increased UCP1 protein stability in a dose-dependent manner without affecting UCP1 transcription, suggesting that miR-210-5p exerts post-transcriptional regulation on UCP1 or broader mitochondrial processes. Although bioinformatic predictions and qPCR analyses indicate that UCP1 is unlikely to be a direct target of miR-210-5p, these results point to the possibility that miR-210-5p may regulate UCP1 indirectly through other mitochondrial-related intermediaries, such as ISCU1/2, which are involved in mitochondrial function [33]. Further studies are necessary to elucidate whether UCP1 is regulated directly by miR-210-5p or through these alternative pathways. Nonetheless, inhibition of miR-210-5p did not lead to significant alterations in oxygen consumption or glycerol concentrations. This lack of metabolic impact may be due to the use of a moderate concentration of the inhibitor, which might have been insufficient to fully stabilize UCP1 expression. The redox environment, influenced by ROS and antioxidant defenses, may also modulate the stability and functionality of UCP1. Future studies should investigate whether miR-210-5p’s effects on UCP1 involve ROS-mediated post-transcriptional mechanisms, further linking redox processes to thermogenesis. However, our findings suggest that miR-210-5p could serve as a potential target for modulating UCP1 expression, warranting further investigation into its metabolic effects in vivo. Additionally, since both UCP1 and HIF-1α are upregulated by NE and CL in brown adipocytes, our findings suggest a scenario in which these two factors may counteract each other in an antagonistic manner or feedback response.

In conclusion, our study highlights that hypoxic conditions adversely affect brown adipogenesis and thermogenic responses, revealing a complex interplay between miR-210-5p and UCP1, while suggesting that hypoxia may hinder adaptive thermogenesis in BAT. Notably, miR-210-5p emerges as a potential candidate for future clinical applications aimed at promoting BAT regeneration, positioning it as a promising therapeutic target for addressing metabolic disorders linked to dysfunctional brown adipocytes. The interplay between hypoxia, ROS, and redox processes emerges as a critical determinant of BAT function. By targeting redox balance and modulating ROS levels, it may be possible to restore BATs thermogenic potential and improve metabolic health. This highlights the need for therapies that address both oxidative stress and hypoxic signaling in BAT regeneration. Our research aims to enhance the understanding of hypoxia’s impact on human health and its role in metabolic regulation.

## 4. Methods and Materials

### 4.1. Primary Cell Collection and Culture

To establish a primary cell culture, brown adipose tissue (BAT) was isolated from 6-week-old male C57BL6/J mice (Janvier Labs, Le Genest-Saint-Isle, France). The tissue was finely minced and enzymatically digested at 37 °C for 30 min in DMEM/F-12 (Sigma-Aldrich, St. Louis, MO, USA), supplemented with 1.2 U/mL Dispase (Roche, Basel, Switzerland), 1 mg/mL collagenase type 2 (Worthington, Lakewood, CA, USA), 15 mg/mL fatty acid-free BSA (Sigma-Aldrich), and 0.1 mg/mL DNase I (Roche), using a shaker. The digestion process was halted by adding fetal bovine serum (FBS, Sigma-Aldrich). The resulting cell suspension was sequentially filtered through 100 µm and 30 µm filters. The stromal vascular fraction (SVF) was collected, plated, and cultured in DMEM/F-12 medium containing 10% *v*/*v* FBS and 1% *v*/*v* penicillin/streptomycin (Thermo Fisher Scientific, Waltham, MA, USA) under standard conditions of 37 °C with 5% CO_2_. Upon reaching approximately 95% confluency, the preadipocytes were induced to differentiate into mature brown adipocytes. From day 0 to day 2 of differentiation, cells were treated with DMEM/F-12 supplemented with 1 mM dexamethasone (Sigma-Aldrich), 340 nM insulin (Sigma-Aldrich), 500 µM isobutylmethylxanthine (Sigma-Aldrich), 2 nM triiodothyronine (Sigma-Aldrich), and 1 µM rosiglitazone (Cayman Chemical, Ann Arbor, MI, USA). From day 2 to day 6, the medium was replaced with DMEM/F-12 supplemented with 10 nM insulin, 2 nM triiodothyronine, and 1 µM rosiglitazone, with media renewal every two days.

### 4.2. Immortalized Cell Culture and Treatment

The WT-1 immortalized mouse brown preadipocyte cell line (generously provided by Brice Emmanueli, University of Copenhagen) was cultured in DMEM Glutamax (Thermo Fisher) supplemented with 10% *v*/*v* FBS and 1% *v*/*v* penicillin/streptomycin. Upon reaching approximately 95% confluency (day 0), differentiation was initiated using an induction medium composed of DMEM Glutamax supplemented with 860 nM insulin, 1 mM dexamethasone, 1 mM triiodothyronine, 1 μM rosiglitazone, 500 mM 3-isobutyl-1-methylxanthine, and 125 mM indomethacin (all from Sigma-Aldrich). After 48 h, this was replaced with differentiation medium containing DMEM Glutamax, 1 mM triiodothyronine, and 1 μM rosiglitazone, with medium refreshed every two days. Full differentiation of the cells was observed within 5–6 days. To achieve specific knockdown of HIF-1α, RNA interference (RNAi) was employed using siRNA targeting HIF-1α (SMARTpool silencing RNA, Dharmacon, Lafayette, CO, USA), while non-targeting siRNA served as a negative control. Reverse transfection was performed with Lipofectamine RNAiMAX transfection reagent (Thermo Fisher), following the manufacturer’s protocol. Transfections were carried out either one day prior to differentiation induction (day −1) or on day 3 of differentiation. Cell treatments were conducted on day 5 of differentiation, with experimental groups treated with 100 μM IOX-2 (Selleck, Houston, TX, USA), 100 μM CoCl_2_ (Sigma-Aldrich), 1 μM CL-316,143 (Tocris, Bristol, UK), or 1 μM norepinephrine (Sigma-Aldrich), compared against controls treated with dimethyl sulfoxide (DMSO, Sigma-Aldrich) or water. For miRNA functional studies and gene regulation analysis, miRCURY LNA™ miRNA inhibitors and mimics targeting miR-210-5p and non-specific siRNA control oligonucleotides (Qiagen, Hilden, Germany) were utilized. All siRNAs and miRNA inhibitors and mimics used, along with their sequences or catalog numbers, are detailed in Appendix A. Transfections of 20 nM miRNA were performed using Lipofectamine 2000 (Invitrogen, Waltham, MA, USA), following established protocols [39].

### 4.3. RNA Isolation and Quantitative PCR (qPCR) Analysis

Total RNA was extracted from cells using the RNeasy Mini Kit (Qiagen, Hilden, Germany), following standard procedures. To ensure DNA-free RNA, on-column DNase digestion was performed using the RNase-free DNase Set (Qiagen) as per the manufacturer’s instructions. cDNA synthesis was conducted using the QuantiTect Reverse Transcription Kit (Qiagen) in accordance with the provided protocol. Quantitative PCR (qPCR) was performed using a QuantStudio 5 Real-Time PCR System (Thermo Fisher Scientific) with the QuantiTect SYBR Green PCR Kit (Qiagen), following the manufacturer’s guidelines. The primers used for amplification are detailed in Appendix A. For miRNA expression analysis, the miRCURY LNA PCR System (Qiagen) was employed to convert RNA into cDNA, and the miRCURY LNA miRNA PCR Assay (Qiagen) was used for quantification. Relative expression levels were normalized to the mean threshold cycle (CT) values of *sno202*, which exhibited minimal variability under treatment conditions. Detailed primer sequences and catalog numbers can be found in Appendix A.

### 4.4. In Vitro Imaging: Immunofluorescence, Confocal Microscopy

The WT-1 immortalized mouse brown preadipocyte cell line was plated onto glass-bottom, 4-well chamber slides (Ibidi, Gräfelfing, Germany) 24 h prior to experimental treatments. After completing the treatments, cells were fixed using 4% paraformaldehyde (PFA) for 15 min and permeabilized with 0.1% Triton X-100 in PBS containing 1% BSA for 30 min at room temperature. Primary antibodies (listed in Appendix A) were applied, and slides were incubated overnight at 4 °C. For imaging, slides were mounted with Prolong^®^ Diamond Antifade Mountant (Thermo Fisher), which included 4′,6-diamidino-2-phenylindole (DAPI) for nuclear staining. Digital images were captured using a Leica DMi8 fluorescence microscope (Leica, Wetzlar, Germany). Higher-resolution localization was achieved through three-dimensional confocal laser scanning microscopy (CLSM), performed with a Leica SP8 3X microscope equipped with a 100xNA1.40 oil immersion objective (Leica). Optical zoom was applied as necessary. Fluorescence excitation was carried out using a UV laser (405 nm) for DAPI and a tunable white light laser for the selective excitation of additional fluorochromes, such as FITC and Star635P. To ensure transparency and reproducibility, all original, unaltered fluorescence images used in the figures are provided in Appendix A.

### 4.5. Protein Isolation and Analysis

Approximately 1.5 million cells per sample were collected in RIPA buffer, consisting of 50 mM Tris (pH 8, Merck, Darmstadt, Germany), 150 mM NaCl (Merck), 0.1% *w*/*v* SDS (Carl Roth, Karlsruhe, Germany), 5 mM EDTA (Merck), and 0.5% *w*/*v* sodium deoxycholate (Sigma-Aldrich), freshly supplemented with a protease inhibitor cocktail (Sigma-Aldrich). Cell lysis was performed using a tissue lyser, followed by two centrifugation steps to remove lipids and debris. Protein concentrations were quantified using the Pierce BCA Assay Kit (Thermo Fisher). For each sample, 15–30 μg of protein was denatured with DTT (Sigma-Aldrich) at 95 °C for 3 min before being loaded onto a NuPage 4–12% Bis-Tris gel (Thermo Fisher) for electrophoretic separation. Following electrophoresis, the proteins were transferred to a 0.2 μm PVDF membrane (Bio-Rad, Hercules, CA, USA) using the Trans-Blot Turbo™ transfer system (Bio-Rad) at 25 V and 1.3 A for 7 min. The membrane was blocked for one hour at room temperature in 5% skim milk. Overnight incubation at 4 °C was carried out with primary antibodies (1:1000 dilution in 5% skim milk), as detailed in Appendix A. The next day, membranes were washed with TBS-T buffer (200 mM Tris, 1.36 mM NaCl, and 0.1% *v*/*v* Tween 20, all from Merck) and then incubated with secondary antibodies (Cell Signaling, Danvers, MA, USA) at a 1:10,000 dilution in TBS-T for 1 h at room temperature. Protein detection was conducted using the SuperSignal West Pico PLUS Chemiluminescent Substrate (Thermo Fisher) and visualized with a ChemiDoc MP Imaging System (Bio-Rad). To maintain transparency and reproducibility, all original and unaltered blot images included in the figures are provided in the Appendix A.

### 4.6. Oil-Red-O (ORO) Staining

Oil Red O (ORO) staining was performed to quantify lipid content. Cells were first rinsed with cold DPBS (Gibco, Carlsbad, CA, USA) and then fixed using a zinc formalin solution (Merck) for 15 min at room temperature. After fixation, cells were washed with 2-propanol (Merck) and allowed to dry. Lipid staining was achieved by incubating the cells with 60% *v*/*v* ORO solution (Sigma) for 10 min at room temperature, followed by 3–4 washes with water to remove excess stain. To visualize lipid accumulation, images of the stained plates were captured. For quantitative analysis, the ORO stain was eluted using 100% 2-propanol, and the optical density (OD) was measured at 500 nm using a plate reader.

### 4.7. Bodipy Staining

Bodipy staining was used to visualize neutral lipid content in cells. Cells were first rinsed with DPBS (Gibco) and subsequently incubated with 2 μM BODIPY 493/503 staining solution (Invitrogen) for 30 min at 37 °C, shielded from light. Following staining, cells were washed with DPBS (Gibco) and fixed with 4% paraformaldehyde (PFA, Thermo Fisher) for 30 min at room temperature. The fixed samples were washed three times with DPBS (Gibco). To prepare for imaging, coverslips were mounted onto glass slides using Prolong^®^ Gold Antifade Reagent with DAPI (Cell Signaling) and allowed to cure overnight at room temperature in the dark. Digital images were captured with a Leica DMi8 fluorescence microscope equipped with a digital camera.

### 4.8. Free Fatty Acid Release Assay

Lipolysis in cell culture supernatants was assessed using the Free Glycerol Reagent (Sigma) and a Glycerol Standard Solution (Sigma). Cell culture medium was collected to quantify free glycerol levels, and the assay was carried out following the manufacturer’s protocol.

### 4.9. Extracellular Flux Analysis (Seahorse)

The oxygen consumption rate (OCR) was evaluated using a Seahorse XFe24 Analyzer (Agilent, Santa Clara, CA, USA), with assays conducted as previously described in the literature. A Seahorse Cell Mito Stress Test (Agilent) was performed following the manufacturer’s protocol, with the inclusion of a CL injection step. Two days prior to the assay, 15,000 adipocytes were seeded into each well. During the assay, cells were sequentially treated with CL (1 μM), oligomycin (1 μM), FCCP (4 μM), and rotenone/antimycin A (both at 0.5 μM). OCR measurements were taken in 3 min intervals. Data normalization was carried out based on total protein content, which was determined using the Pierce BCA Assay Kit (Thermo Fisher) following the manufacturer’s guidelines. CL-induced respiration was calculated as the difference between the maximum OCR induced by CL and the maximum baseline OCR. Coupled respiration was determined by subtracting the OCR after oligomycin injection from the baseline OCR. Uncoupled respiration was defined as the OCR difference between the post-oligomycin and post-Rot/AA injection values. Maximum respiration was determined by subtracting the minimum OCR (after Rot/AA injection) from the maximum OCR (after FCCP injection). The raw data were processed and analyzed using Wave Controller 2.6.1 (Agilent).

### 4.10. Data Analysis and Visualization

Data visualization was performed in Excel, with statistical analysis carried out using GraphPad Prism version 8.0.2. Results are expressed as mean ± SD (n = 3). A multiple Student’s *t*-test, followed by Bonferroni post-hoc analysis, was applied to compare two groups with a single variable. For comparisons among three groups with one variable, a one-way ANOVA with Tukey’s post-hoc test was employed. Two-way ANOVA with Tukey’s post-hoc test was used to analyze differences between four groups involving two variables, while three-way ANOVA with Dunnett’s post-hoc test was utilized for comparisons among more than four groups with more than two variables. Statistical significance was defined as * *p* < 0.05; ** *p* < 0.01; *** *p* < 0.001.

## 5. Conclusions

This study demonstrates that hypoxic conditions impair brown adipocyte differentiation and thermogenic responses, with miR-210-5p playing a significant regulatory role. We observed that hypoxia-induced stabilization of HIF-1α upregulates miR-210-5p, which affects brown adipocyte differentiation and lipid storage. Under normoxic conditions, miR-210-5p enhances differentiation, but its effects are insufficient to rescue the inhibition of brown adipocyte differentiation under hypoxic conditions. Although miR-210-5p does not directly regulate UCP1 expression, our results suggest that it may influence UCP1 protein stability through post-transcriptional mechanisms involving mitochondrial-related intermediaries. Despite the lack of significant changes in oxygen consumption or glycerol release upon miR-210-5p inhibition, our findings highlight the potential of miR-210-5p as a target for modulating brown adipocyte function. This study underscores the importance of understanding the role of miRNAs in the regulation of brown adipose tissue under hypoxic conditions and provides insights into how oxidative stress, redox signaling, and miR-210-5p may collectively impact adipocyte thermogenesis. Future work will be essential to determine the precise mechanisms by which miR-210-5p modulates thermogenesis and explore its therapeutic potential for metabolic diseases associated with dysfunctional brown adipocytes.

## Figures and Tables

**Figure 1 ijms-26-00117-f001:**
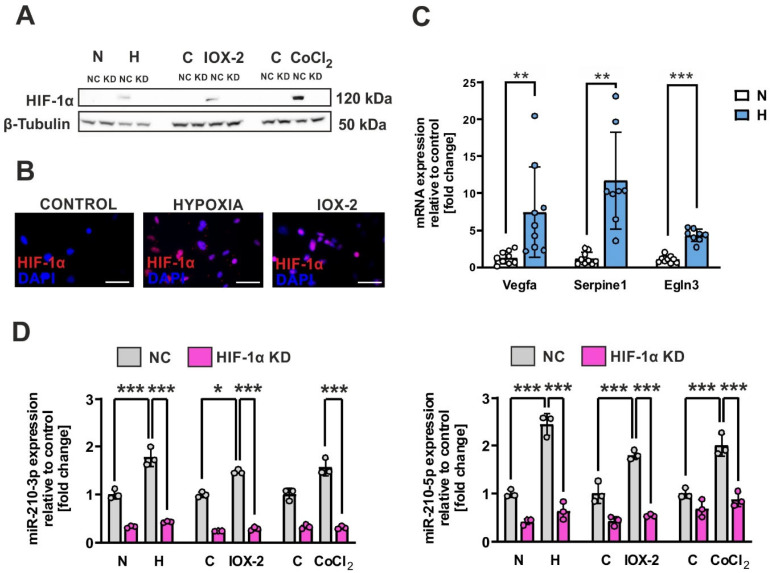
Hypoxia and HIF-1α stabilization regulate miR-210 expression in brown adipocytes. (**A**) Western blotting analysis of HIF-1α protein expression in immortalized brown adipocytes exposed to normoxic (N) or hypoxic conditions (H) or treated with IOX2 (100 µM) and CoCl_2_ (100 µM) under normoxic conditions for 24 h. (**B**) Subcellular localization of HIF-1α was examined by immunocytochemistry analysis under normoxic conditions (Control), hypoxic conditions for 12 h, or by incubation with IOX-2 (100 µM) for 3 h. Scale bars 100 µm. (**C**) qPCR analysis of HIF-1α target genes after incubation of brown adipocytes under normoxic (N) or hypoxic conditions (H) for 12 h. (**D**) Immortalized brown adipocytes were transfected with either negative control siRNA (NC) or HIF-1α siRNA (KD) 2 days prior to the assay. Subsequently, the cells were exposed to normoxic (N) or hypoxic conditions (H) or treated with IOX2 (100 µM) and CoCl_2_ (100 µM) under normoxic conditions for 24 h. After 24 h, RNA was collected and subjected to quantification of miR-210-3p and -5p expression using qPCR analysis. The miRNA levels were normalized to sno202. The data represent the mean ± SD (n = 3); Statistical significance is when * *p* < 0.05, ** *p* < 0.01, *** *p* < 0.001.

**Figure 2 ijms-26-00117-f002:**
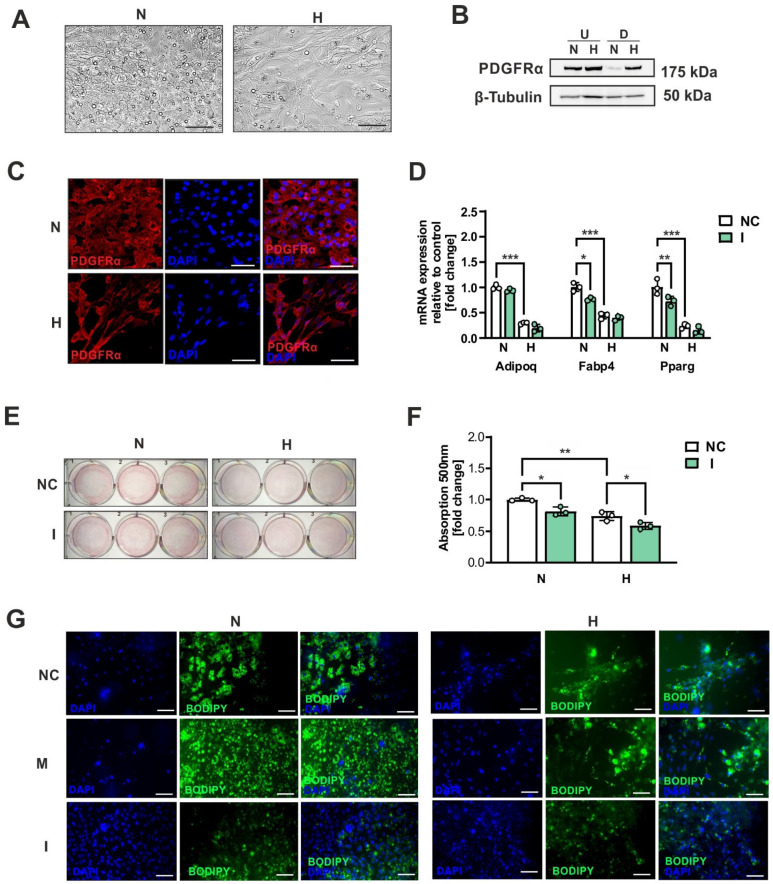
Hypoxia impairs brown adipocyte differentiation. (**A**) Microscopic analysis of immortalized brown adipocytes differentiated under normoxic (N) or hypoxic conditions (H). (**B**) Immunoblot analysis of PDGFRα and β-Tubulin in undifferentiated (U) and differentiated brown adipocytes (D) cultured under normoxic (N) or hypoxic conditions (H) for 24 h. (**C**) Immunofluorescence analysis of PDGFRα (red) and DAPI (blue) of immortalized brown adipocytes differentiated under normoxic (N) or hypoxic conditions (H). (**D**) qPCR for adipogenic markers of immortalized brown adipocytes transfected with either negative control siRNA (NC) or LNA miR-210-5p inhibitor (I) and differentiated under normoxic (N) or hypoxic conditions (H) for 5 days. (**E**) Cell staining with Oil Red O for adipogenic differentiation after 5 days of differentiation under normoxic (N) or hypoxic (H) conditions is shown by representative microscopic images of stained cellular monolayers. (**F**) Oil Red O spectrophotometry by Tecan plate reader and stain recovery after extraction from the cells. (**G**) Bodipy staining analysis of immortalized brown adipocytes transfected with either negative control miRNA (NC) or LNA miR-210-5p-mimic (M) and -inhibitor (I) after differentiation under normoxic (N) or hypoxic conditions (H). Data shown represent the mean ± SD of triplicate measurements (n = 3). Stadistical significance when * *p* < 0.05, ** *p* < 0.01, *** *p* < 0.001. Scale bars 100 µm.

**Figure 3 ijms-26-00117-f003:**
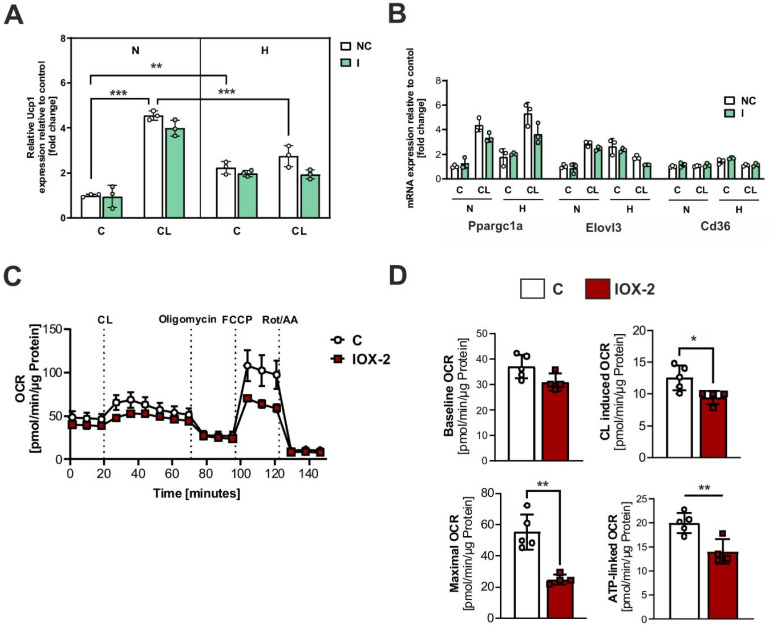
Hypoxia impairs the thermogenic capacity of brown adipocytes. (**A**) qPCR analysis of *Ucp-1* in primary brown adipocytes transfected with either control miRNA (NC) or LNA miR-210-5p inhibitor (I) and treated under normoxic or hypoxic conditions with or without CL (1 µM) for a duration of 4 h. (**B**) qPCR of *Ppargc1a*, *Elovl3*, and *Cd36* in primary brown adipocytes transfected with either negative control miRNA (NC) or LNA miR-210-5p inhibitor (I) and treated under normoxic or hypoxic conditions with or without CL (1 µM) for a duration of 4 h. (**C**,**D**) Oxygen consumption rates in negative control miRNA (NC) and LNA miR-210-5p inhibitor (I) transfected brown adipocytes in the presence or absence of IOX-2 (100 µM) for 2 h and normalized to protein content. Data shown represent the mean ± SD of triplicate measurements (n = 5). Stadistical significance when * *p* < 0.05; ** *p* < 0.01; *** *p* < 0.001.

**Figure 4 ijms-26-00117-f004:**
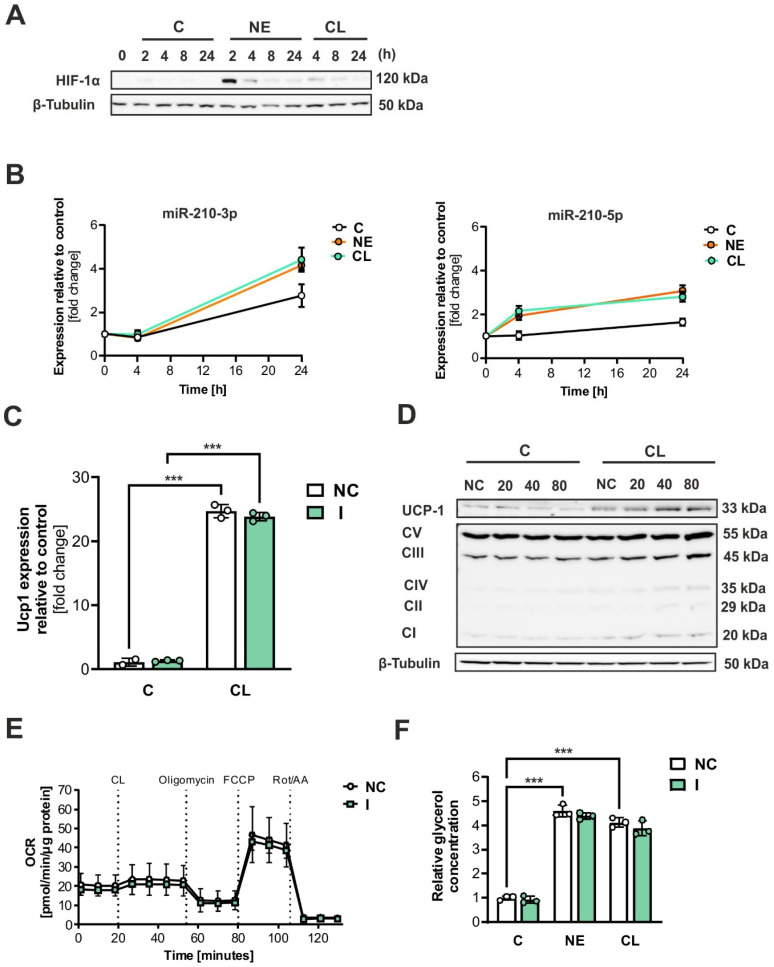
Adrenergic stimuli activate HIF-1α signaling, leading to an increase in miR-210 expression. (**A**) Immunoblot analysis of HIF-1α in immortalized brown adipocytes after exposure to NE (1 µM) and CL (1 µM) for 2, 4, 8, or 24 hours. (**B**) qPCR analysis of miR-210-3p and -5p expression after 4 and 24 h incubation with NE (1 µM) and CL (1 µM). (**C**) qPCR analysis of *Ucp-1.* (**D**) Immunoblot analysis of UCP-1, Electron Transport Chain Complexes, and β-tubulin in immortalized brown adipocytes after transfection with (0, 20, 40, or 80) nM of miR-210-5p inhibitor with or without CL (1 µM) stimulation. (**E**) Oxygen consumption rate of transfected adipocytes (**F**) Quantification of glycerol concentration of transfected brown adipocytes 6 h after incubation with either NE or CL (1 µM). Data shown represent the mean ± SD of triplicate measurements (n = 3). Stadistical significance when *** *p* < 0.001.

## Data Availability

The raw data supporting the conclusions of this article will be made available by the authors without undue reservation.

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
