# Peer review of "Hypoxia Regulates Brown Adipocyte Differentiation and Stimulates miR-210 by HIF-1α"

_ijms, 2024, doi:10.3390/ijms26010117_

Round 1

Reviewer 1 Report

Comments and Suggestions for Authors

This manuscript investigated the effect of hypoxia and adrenergic stimulation on adipocyte differentiation and thermogenesis, especially focused on the role of “master hypoxamiR” miR-210 in these processes. The section on hypoxia treatment and adrenergic stimulation is very clear and well-supported by the data. However, the role of miR-210 has not been clearly determined.  

For adipocyte differentiation part, In Figure 1, hypoxia was shown to induce the expression of miR-210. Figure 2, Hypoxia impairs brown adipocyte differentiation; however, KD of miR-210 suppressed the brown adipocyte differentiation and lipid storage capacity, as shown in Figure 2E to G. Although the authors discussed this part in Line 404-405, the role of miR-210 is still not clear under hypoxia. It may suggest the miR-210 needs to be maintained at a specific level for function. Rescue assay is necessary to validate if miR-210 promote or inhibit the adipocyte differentiation.

Similarly, it is hard to see a clear relationship between miR-210-5p and UCP-1 expression. Line 340, the authors claimed, “a dose-dependent increase in UCP-1 protein levels was observed under CL stimulation”. However, from the WB in Figure 4D, the UCP-1 band of 80nM is slight lower than that of 40nM. This needs more experiments for quantification. From qPCR and bioinformatic prediction, UCP-1 may not be the downstream target of mi-210-5p. Many studies show that miR-210-5p is known to influence mitochondrial function and metabolism by targeting ISCU1/2, which may indirectly affect the UCP-1 function. Did the author test that? This needs to be included in discussion.

Minor :

1.     Line75 “referred”

2.     Line 107, "100mM filter and 30mM filter" , the unit are not correct. It should be 100um or 30um cell strainer.

3.     Line, 318 to 326, "the protein level of HIF-1" data and figures are  missing for this part. Figure 3C and D are also not mentioned.

4.     Line 358, n is not 3, as shown in scatter dot plot in Figure 3D.

5.     Line 356there is no “negative control miRNA and LNP miR-210 inhibitor” shown in C and D.

6.     Some experiments used primary mouse brown adipocytes; some used immortalized cell line. Include the cell information in figure legends.

Author Response

We would like to express our sincere gratitude to the reviewers for their insightful and constructive comments, which have significantly improved the quality of our manuscript. In addition to addressing the specific changes requested, we have updated the entire Materials and Methods section to provide greater clarity. Furthermore, we have added an additional conclusion section highlighting the key findings and implications of our study. We believe these revisions have enhanced the manuscript, and we are confident that the revised version better communicates the significance of our research.

Reviewer #1

This manuscript investigated the effect of hypoxia and adrenergic stimulation on adipocyte differentiation and thermogenesis, especially focused on the role of “master hypoxamiR” miR-210 in these processes.

Major points:

  1. The section on hypoxia treatment and adrenergic stimulation is very clear and well-supported by the data. However, the role of miR-210 has not been clearly determined.

RESPONSE: We thank the reviewer for the positive feedback regarding the clarity and data supporting the hypoxia treatment and adrenergic stimulation sections. Regarding the role of miR-210, we have now clarified its dual function in the revised manuscript. Additionally, we performed an additional assay using miR-210 mimics, which demonstrated that miR-210 supports brown adipocyte differentiation. These findings are included in the updated manuscript (new Fig.2G).

  1. For adipocyte differentiation part, In Figure 1, hypoxia was shown to induce the expression of miR-210. Figure 2, Hypoxia impairs brown adipocyte differentiation; however, KD of miR-210 suppressed the brown adipocyte differentiation and lipid storage capacity, as shown in Figure 2E to G. Although the authors discussed this part in Line 404-405, the role of miR-210 is still not clear under hypoxia. It may suggest the miR-210 needs to be maintained at a specific level for function. Rescue assay is necessary to validate if miR-210 promote or inhibit the adipocyte differentiation.

RESPONSE: We thank the reviewer for this insightful comment regarding the role of miR-210 in brown adipocyte differentiation under hypoxia. In response to the reviewer’s suggestion, we performed an assay overexpressing miR-210-5p using LNA-mimics to further elucidate its role and have included this data on the main manuscript (new Fig.2 G). Our results demonstrated that overexpressing miR-210 enhanced brown differentiation under normoxic conditions and partially restored brown adipocyte differentiation and lipid storage capacity under hypoxic conditions. This indicates that miR-210 indeed plays a supportive role in adipocyte differentiation. Based on these new findings, we have updated the manuscript (Lines 26-28, 145-147, 177-179) to reflect the role of miR-210 in brown adipocyte differentiation. Our results underscore the complex role of miR-210 in adipogenesis, as its functions depend on both its expression levels and the cellular environment (e.g., normoxia vs. hypoxia). This dual role is discussed in greater detail in the revised manuscript to address the reviewer’s concern and provide a more comprehensive understanding of its regulatory potential.

  1. Similarly, it is hard to see a clear relationship between miR-210-5p and UCP-1 expression. Line 340, the authors claimed, “a dose-dependent increase in UCP-1 protein levels was observed under CL stimulation”. However, from the WB in Figure 4D, the UCP-1 band of 80nM is slight lower than that of 40nM. This needs more experiments for quantification. From qPCR and bioinformatic prediction, UCP-1 may not be the downstream target of mi-210-5p. Many studies show that miR-210-5p is known to influence mitochondrial function and metabolism by targeting ISCU1/2, which may indirectly affect the UCP-1 function. Did the author test that? This needs to be included in discussion.

RESPONSE: We appreciate the reviewer’s comments regarding the relationship between miR-210-5p and UCP-1 expression. We acknowledge that while our Western blot in Figure 4D shows a slight variability between 40 nM and 80 nM treatment, the overall trend of UCP-1 protein increasing with miR-210-5p inhibitor under adrenergic stimulation is consistent. To address the reviewer’s concern, we have revised the manuscript to clarify this observation and emphasize the semi-quantitative nature of Western blot analysis, which may explain minor variability in band intensity (Lines 211-217). Regarding UCP-1 as a potential downstream target of miR-210-5p, we agree that qPCR and bioinformatic predictions indicate that UCP-1 is unlikely to be a direct target. Instead, as the reviewer highlights, miR-210-5p is known to regulate mitochondrial function through targets such as ISCU1/2, which could indirectly affect UCP-1 expression. Additionally, these effects may be mediated by miR-210 influencing other intermediaries involved in UCP-1 stability. While we have not specifically tested these mechanisms, we have incorporated a discussion of these possibilities, supported by relevant literature, into the revised manuscript (Lines 321-329). We thank the reviewer for highlighting this important point, which has allowed us to provide greater context and depth to our discussion.

Minor points:

  1. Line75, “referred”

RESPONSE: This word has now been corrected.

  1. Line 107, "100mM filter and 30mM filter", the unit are not correct. It should be 100um or 30um cell strainer.

RESPONSE: Thank you for carefully proofreading our manuscript.

  1. Line, 318 to 326, "the protein level of HIF-1" data and figures are missing for this part. Figure 3C and D are also not mentioned.

RESPONSE: We thank the reviewer for pointing this out. We apologize for the oversight. We have added referenced these figures in the relevant section (Lines 192–195) to ensure proper alignment with the results presented.

  1. Line 358, n is not 3, as shown in scatter dot plot in Figure 3D.

RESPONSE: We thank the reviewer for pointing out this discrepancy. We apologize for the error in reporting the sample size. The correct sample size for the data shown in Figure 3D is indeed not 3, as stated in Line 358. We have updated the manuscript to reflect the correct sample size and have revised Figure 3D accordingly.

  1. Line 356, there is no “negative control miRNA and LNP miR-210 inhibitor” shown in C & D.

RESPONSE: We thank the reviewer for pointing this out. We apologize for the oversight. We have now corrected and have updated the manuscript accordingly to reflect these changes.

  1. Some experiments used primary mouse brown adipocytes; some used immortalized cell line. Include the cell information in figure legends.

RESPONSE: We thank the reviewer for this helpful suggestion. We have now included detailed information regarding the cell types used in the experiments in the figure legends, specifying whether primary mouse brown adipocytes or the immortalized cell line was employed. This clarification has been added to ensure transparency and reproducibility.

Reviewer 2 Report

Comments and Suggestions for Authors

This study aims to investigate the influence of hypoxia and the associated hypoxamiR, miR-210, on brown adipose tissue (BAT) differentiation and thermogenesis. They showed that hypoxia and adrenergic stimulation upregulated miR-210, which impacted brown adipocyte differentiation and thermogenesis.

Comments

-Abstract and Introduction

The Introduction and abstract are well-written. 

- Materials, Methods, and Results

- Abbreviations should be given only after their full names are provided for the first time and should be consistently used thereafter (e.g., UCP1)

- It is not clear how hypoxic condition (1% O2) was induced. Is it conducted through a hypoxic chamber or a hypoxic incubator?

- On what basis CoCl2 at 600 µM was used? Does CoCl2 at 600 µM induce cytotoxicity? Is MTT assay carried out on brown adipocytes using serial dilution of CoCl2?

- What is the reason for using different hypoxia inducers (HIF-1α activators and stabilizers): (1% O2), IOX2, or CoCl2?

- In the introduction, the authors stated, “However, it remains unclear whether the regulation of miR-210 during hypoxia is in the form of miR-210-5p, miR-210-3p or both concomitantly”. In the results, the authors stated, “Given the pronounced induction of miR-210-5p under these conditions, further studies focused on this strand”. However, results showed that both miR-210-5p and miR-210-3p were significantly expressed in brown adipocytes exposed to normoxic (N) or hypoxic conditions (H) or treated with IOX2 (100 µM) and CoCl2 (600 µM) under normoxic conditions for 24 hours (Figure 1D). Moreover, qPCR analysis of miR-210-3p and -5p expression after 4- and 24-hours incubation with NE (1µM) and CL (1µM) showed pronounced induction of miR-210-3p in brown adipocytes after 24 hours (Figure 4B).  So, why was the only sequence of miR-210-5p Inhibitor used? Both sequences should be tested.

In Figure 1C, the authors need to review the statistical analysis of the Serpine 1HIF-1α target gene as it showed a significant (***p < 0.001) increase under hypoxic conditions despite the high SD.

- In hypoxia impairs brown adipocyte differentiation experiments; why was hypoxia induced using 1% O2? and IOX2, or CoCl2 was not used. Similarly, in hypoxia impairs the thermogenic capacity of brown adipocyte experiments; 1% O2 and IOX2 were used, whereas CoCl2 was not used.

- In hypoxia impairs brown adipocyte differentiation experiments, authors transfected the cells with negative control miRNA (NC) or LNA miR-210-5p inhibitor to investigate the role of miR-210-5p in the adipocyte differentiation markers Adipoq, Fabp and Pparg. Results showed that the adipocyte differentiation markers Adipoq, Fabp, and Pparg expression were significantly (***p < 0.001) downregulated in cells transfected with negative control miRNA (NC) under the hypoxic conditions compared with normoxic conditions (Figure 2B). However, under the hypoxia, the nonsignificant decrease of these markers in cells transfected with LNA miR-210-5p inhibitor compared with cells transfected with negative control miRNA (NC) conditions makes the role of miR-210-5p in adipocyte differentiation markers under hypoxic conditions questionable.

In addition, in hypoxia impairs the thermogenic capacity of brown adipocyte experiments, no significant effect was observed in the UCP1 expression level in primary brown adipocytes transfected with either NC miRNA or LNA miR-210-5p inhibitor and treated with CL under normoxic or hypoxic conditions (Figure 3A). This may weaken the claimed role of miR-210-5p in the thermogenic capacity of brown adipocytes.

In Figure 4A, immunoblot analysis of HIF-1α in brown adipocytes after exposure to CL (1µM), the 2-hour time point is missing.

Compared with norepinephrine (NE), the β3-adrenergic agonist (CL316243) is a highly potent selective β3-adrenoceptor agonist but is an extremely poor to β1/2- receptors, > 10000-fold selective over β1 and β2 receptors (Tocris). Given that β3-adrenoceptor is predominantly expressed in brown adipocytes, what is the reason behind using both NE and CL as adrenergic stimuli?

Based on this study's results, the abstract, discussion, and conclusion should accurately describe the role of miR-210-5p.

-Discussion

The authors stated,” Surprisingly, the miR-210-5p inhibitor significantly increased UCP1 protein stability in a dose-dependent manner, without affecting UCP1 transcription, suggesting that miR-210-5p exerts post-transcriptional regulation on UCP1 or entire mitochondria”. Can authors elaborate more on how this could happen?

Author Response

We would like to express our sincere gratitude to the reviewers for their insightful and constructive comments, which have significantly improved the quality of our manuscript. In addition to addressing the specific changes requested, we have updated the entire Materials and Methods section to provide greater clarity. Furthermore, we have added an additional conclusion section highlighting the key findings and implications of our study. We believe these revisions have enhanced the manuscript, and we are confident that the revised version better communicates the significance of our research.

Reviewer #2

This study aims to investigate the influence of hypoxia and the associated hypoxamiR, miR-210, on brown adipose tissue (BAT) differentiation and thermogenesis. They showed that hypoxia and adrenergic stimulation upregulated miR-210, which impacted brown adipocyte differentiation and thermogenesis. Comments:

  1. Abstract and Introduction: The Introduction and abstract are well-written.

RESPONSE: We thank the reviewer for their positive feedback regarding the abstract and introduction. We are pleased that these sections were well received and appreciate your encouraging comments.

  1. Materials, Methods, and Results: Abbreviations should be given only after their full names are provided for the first time and should be consistently used thereafter (e.g., UCP1)

RESPONSE: Thank you for pointing this out. We have now revised the manuscript to ensure that all abbreviations, including UCP1, are introduced with their full names upon first mention, followed by the appropriate abbreviations in subsequent references.

  1. It is not clear how hypoxic condition (1% O2) was induced. Is it conducted through a hypoxic chamber or a hypoxic incubator?

RESPONSE: We thank the reviewer for this question. Hypoxic conditions (1% O₂) were induced using a hypoxic incubator, which precisely controls the oxygen concentration and ensures consistent exposure to hypoxia. We have now clarified this point in the revised manuscript to provide more detailed information on the experimental setup. (Line 106)

  1. On what basis CoCl2 at 600 µM was used? Does CoCl2 at 600 µM induce cytotoxicity? Is MTT assay carried out on brown adipocytes using serial dilution of CoCl2?

RESPONSE: We thank the reviewer for pointing out the discrepancy. We apologize for the error in the initial manuscript. The correct concentration of CoCl₂ used in our experiments was 100 µM, not 600 µM as previously stated. This concentration was chosen based on previous studies in the literature, where 100 µM CoCl₂ has been shown to effectively stabilize HIF-1α without inducing significant cytotoxicity. We have corrected this information in the revised manuscript and updated the relevant sections accordingly.

  1. What is the reason for using different hypoxia inducers (HIF-1α activators and stabilizers): (1% O2), IOX2, or CoCl2?

RESPONSE: We thank the reviewer for this question regarding the use of different hypoxia inducers (1% O₂, IOX2, and CoCl₂). Each inducer was selected to explore complementary aspects of hypoxia and HIF-1α activation. Specifically, 1% O₂ represents physiological hypoxia, directly mimicking reduced oxygen availability and activating HIF-1α through oxygen-dependent pathways. IOX2, a specific prolyl hydroxylase (PHD) inhibitor, stabilizes HIF-1α under normoxic conditions, allowing us to isolate its effects independently of oxygen depletion. CoCl₂, another PHD inhibitor, chemically mimics hypoxia by stabilizing HIF-1α and can also trigger additional transcriptional responses that provide broader insights into hypoxia-mimetic conditions. This multi-approach strategy ensures that our conclusions are robust and not reliant on a single method of hypoxia induction. It also allows us to differentiate between effects caused by reduced oxygen tension and those driven specifically by HIF-1α stabilization. We have clarified this rationale in the revised manuscript (Lines 108-112).

  1. In the introduction, the authors stated, “However, it remains unclear whether the regulation of miR-210 during hypoxia is in the form of miR-210-5p, miR-210-3p or both concomitantly”. In the results, the authors stated, “Given the pronounced induction of miR-210-5p under these conditions, further studies focused on this strand”. However, results showed that both miR-210-5p and miR-210-3p were significantly expressed in brown adipocytes exposed to normoxic (N) or hypoxic conditions (H) or treated with IOX2 (100 µM) and CoCl2 (100 µM) under normoxic conditions for 24 hours (Figure 1D). Moreover, qPCR analysis of miR-210-3p and -5p expression after 4- and 24-hours incubation with NE (1µM) and CL (1µM) showed pronounced induction of miR-210-3p in brown adipocytes after 24 hours (Figure 4B). So, why was the only sequence of miR-210-5p Inhibitor used? Both sequences should be tested.

RESPONSE: We thank the reviewer for raising this important point. While both miR-210-3p and miR-210-5p are upregulated in brown adipocytes under hypoxic conditions (Figure 1D), we specifically focused on miR-210-5p in our inhibitor studies. While miR-210-3p has been more extensively studied and is considered the primary functional strand in many cellular processes, including hypoxia-related pathways, the 5p strand has also been shown to play important roles in certain contexts. Given the pronounced induction of miR-210-5p in our system and its emerging role in regulating adipocyte differentiation, we chose to investigate the 5p strand in this study. We acknowledge that miR-210-3p could also have significant effects, and we agree that testing both strands could provide a more comprehensive understanding. This will be considered in future investigations. We have now clarified this rationale in the revised manuscript (Lines 262-269).

  1. In Figure 1C, the authors need to review the statistical analysis of the Serpine 1HIF-1α target gene as it showed a significant (***p < 0.001) increase under hypoxic conditions despite the high SD.

RESPONSE: We thank the reviewer for their observation regarding the statistical analysis of the Serpine 1. This error has been corrected in the actual manuscript.

  1. In hypoxia impairs brown adipocyte differentiation experiments; why was hypoxia induced using 1% O2? and IOX2, or CoCl2 was not used. Similarly, in hypoxia impairs the thermogenic capacity of brown adipocyte experiments; 1% O2 and IOX2 were used, whereas CoCl2 was not used.

RESPONSE: We thank the reviewer for the question regarding our choice of hypoxia-inducing methods. In the experiments examining how hypoxia impairs brown adipocyte differentiation, we used 1% O₂ to create a physiologically relevant hypoxic environment that closely mimics in vivo conditions. Chemical mimetics of hypoxia, such as IOX2 and CoCl₂, were not used in these experiments to avoid potential off-target effects or artifacts introduced by chemical treatments. In contrast, in the experiments investigating the impact of hypoxia on the thermogenic capacity of brown adipocytes, we employed both 1% O₂ and IOX2. This dual approach allowed us to validate the role of hypoxia-inducible factor (HIF) stabilization in thermogenic impairment, with IOX2 specifically targeting the prolyl hydroxylase enzymes that regulate HIF stability. CoCl₂ was not included in these experiments due to its broader nonspecific effects on cellular processes.

  1. In hypoxia impairs brown adipocyte differentiation experiments, authors transfected the cells with negative control miRNA (NC) or LNA miR-210-5p inhibitor to investigate the role of miR-210-5p in the adipocyte differentiation markers Adipoq, Fabp and Pparg. Results showed that the adipocyte differentiation markers Adipoq, Fabp, and Pparg expression were significantly (***p < 0.001) downregulated in cells transfected with negative control miRNA (NC) under the hypoxic conditions compared with normoxic conditions (Figure 2B). However, under the hypoxia, the nonsignificant decrease of these markers in cells transfected with LNA miR-210-5p inhibitor compared with cells transfected with negative control miRNA (NC) conditions makes the role of miR-210-5p in adipocyte differentiation markers under hypoxic conditions questionable.

RESPONSE: We thank the reviewer for their thoughtful comment. To address this concern and provide further insights, we conducted additional experiments overexpressing miR-210-5p using LNA mimics, as detailed in the revised manuscript (new Fig. 2G). Our findings demonstrated that miR-210-5p overexpression significantly enhanced brown adipocyte differentiation under normoxic conditions and partially rescued brown adipocyte differentiation and lipid storage capacity under hypoxic conditions. These new results indicate that miR-210-5p indeed plays a supportive role in promoting adipocyte differentiation, particularly under hypoxic stress. To address the apparent insignificant changes observed with the LNA miR-210-5p inhibitor in the initial experiments, we now interpret these findings as reflective of the dual and context-dependent role of miR-210-5p in adipogenesis. This role appears to depend on its expression levels and the prevailing cellular environment, such as normoxia versus hypoxia. We have updated the manuscript (Lines 26-28, 145-147, 177-179) to reflect these findings and provide a more nuanced understanding of miR-210-5p’s regulatory potential in brown adipocyte differentiation. We hope this additional data and the expanded interpretation address the reviewer’s concerns comprehensively.

  1. In addition, in hypoxia impairs the thermogenic capacity of brown adipocyte experiments, no significant effect was observed in the UCP1 expression level in primary brown adipocytes transfected with either NC miRNA or LNA miR-210-5p inhibitor and treated with CL under normoxic or hypoxic conditions (Figure 3A). This may weaken the claimed role of miR-210-5p in the thermogenic capacity of brown adipocytes.

RESPONSE: We thank the reviewer for raising this important point. While we did not observe significant changes in UCP1 expression levels in primary brown adipocytes transfected with either NC miRNA or LNA miR-210-5p inhibitor under normoxic or hypoxic conditions (Figure 3A), we would like to emphasize that post-transcriptional regulation could still play a critical role in the thermogenic capacity of brown adipocytes. The lack of a significant change at the transcriptional level does not preclude the possibility that miR-210-5p regulates UCP1 protein stability, translation efficiency, or other post-transcriptional mechanisms that influence UCP1 function. Indeed, miR-210-5p may modulate the stability or activity of UCP1 through factors such as protein degradation pathways (e.g., proteasomal or autophagic pathways) or through translational control, which may not be immediately apparent at the mRNA level. We have revised the manuscript to clarify that the regulation of thermogenic capacity may involve both transcriptional and post-transcriptional mechanisms, and that miR-210-5p's effects may be more prominent at the protein level rather than at the mRNA level (Lines 321-329).

  1. In Figure 4A, immunoblot analysis of HIF-1α in brown adipocytes after exposure to CL (1µM), the 2-hour time point is missing.

RESPONSE: Thank you for this comment. As we observed higher increase in HIF-1α expression at the 4-hour time point, we have insight into the temporal dynamics of HIF-1α under CL exposure.

  1. Compared with norepinephrine (NE), the β3-adrenergic agonist (CL316243) is a highly potent selective β3-adrenoceptor agonist but is an extremely poor to β1/2- receptors, > 10000-fold selective over β1 and β2 receptors (Tocris). Given that β3-adrenoceptor is predominantly expressed in brown adipocytes, what is the reason behind using both NE and CL as adrenergic stimuli?

RESPONSE: We thank the reviewer for this important question regarding the use of both norepinephrine (NE) and CL316243 as adrenergic stimuli in our study. NE was used as a physiological agonist to mimic cold-induced adrenergic stimulation, as it activates all adrenergic receptor subtypes (β1, β2, and β3), which reflects the natural signaling in vivo. In contrast, CL316243 was included as a highly selective β3-adrenergic agonist to specifically target β3-adrenoceptors, which are predominantly expressed in brown adipocytes. By including both NE and CL, we aimed to explore the differential responses of brown adipocytes to general adrenergic stimulation versus β3-specific stimulation, allowing us to assess both physiological and receptor-specific effects. We have now clarified this rationale in the Methods and Discussion sections of the revised manuscript (Lines 199-203).

  1. Based on this study's results, the abstract, discussion, and conclusion should accurately describe the role of miR-210-5p.

RESPONSE: Thank you for your valuable comment. We completely agree that the abstract, discussion, and conclusion should accurately reflect the role of miR-210-5p, as described in our study’s findings. Based on the data, we now emphasize that miR-210-5p plays a multifaceted role in brown adipocyte differentiation and thermogenesis, with its effects being likely mediated through both transcriptional and post-transcriptional mechanisms.

  1. Discussion: The authors stated,” Surprisingly, the miR-210-5p inhibitor significantly increased UCP1 protein stability in a dose-dependent manner, without affecting UCP1 transcription, suggesting that miR-210-5p exerts post-transcriptional regulation on UCP1 or entire mitochondria”. Can authors elaborate more on how this could happen?

RESPONSE: We appreciate the reviewer’s request for further clarification regarding the observation that the miR-210-5p inhibitor significantly increased UCP1 protein stability without affecting its transcription. This finding suggests that miR-210-5p may regulate UCP1 through a post-transcriptional mechanism. One possible explanation is that miR-210-5p could influence the stability of UCP1 protein by modulating factors involved in its degradation, such as the ubiquitin-proteasome system or autophagic pathways. miR-210-5p may target mRNA-binding proteins or other translational regulators that affect UCP1’s protein half-life, thus stabilizing it under conditions where miR-210-5p levels are reduced. Additionally, miR-210-5p might regulate mitochondrial dynamics or quality control processes, which could indirectly affect UCP1 protein turnover. These hypotheses are consistent with the broader role of miR-210 in regulating mitochondrial function and homeostasis, as documented in previous studies. We have now expanded the discussion in the revised manuscript to incorporate these potential mechanisms (Lines 321-329). We hope this additional explanation addresses the reviewer’s concern.

Round 2

Reviewer 1 Report

Comments and Suggestions for Authors

The authors have addressed the majority of my previous comments.  I do not have further request. 

Minor error: Line  362,  it is supposed to be "100 µm and 30 µm filters", instead of " 100 µM and 30 µM filters".  µM is the molar concentration. 

Reviewer 2 Report

Comments and Suggestions for Authors

I have no more comments.